# Impact of Soil on the Penetration of Prefabricated Spherical Fragments and Its Protective Effect

Zhenning Wang, Jianping Yin *, Zhijun Wang, Xudong Li and Jianya Yi

College of Mechanical and Electrical Engineering, North University of China, Taiyuan 030051, China;
wzn199603@163.com (Z.W.); wzj@nuc.edu.cn (Z.W.); lxdscibizhong@126.com (X.L.); yijianya513@126.com (J.Y.)
* Correspondence: yjp123@nuc.edu.cn

**Abstract:** As a natural environmental medium, soil has a wide range of sources and is often used as a material for building houses. It can also be used to construct simple protective structures in actual battlefield environments. In order to study the protective effect of sand on prefabricated fragments in practical situations, the authors conducted experiments on spherical tungsten alloy prefabricated fragments penetrating into sand using sand as the target medium. Natural sand (naturally generated rock particles with a particle size less than 4.75 mm) was selected as the sand, and the initial velocities of the fragments were 689 m/s~1761 m/s (fragment diameter 6 mm) and 596 m/s~1325 m/s (fragment diameter 11 mm), respectively. Based on the residual velocity of fragments and experimental phenomena in experimental data, the authors used LS-DYNA software to numerically simulate and compare the residual velocity of fragments and obtained a numerical model for sandy soil media and a calculation formula for the residual velocity of spherical fragments. Based on modeling equivalence in practical environments, the authors studied the impact of different attack angles on fragment trajectory deviation and found that fragments have the highest deviation values at 20° and 70° attack angles. They also analyzed the soil boundary effect within a small scale range, and the effect was significant when the vertical distance was less than 40 mm. The penetration resistance of different thicknesses of soil was calculated, and the maximum soil thickness under prefabricated fragment penetration at different speeds was obtained. The effective protective size of the soil under actual conditions was about 550 mm. This experiment provides a reference for the construction parameters of simple soil defense structures in actual environments.

**Keywords:** soil; prefabricated fragments; protective fortifications; residual speed

## 1. Introduction

As a natural medium in nature, soil constitutes various environmental types on land and has played an important role in human society from ancient times to the present. In earlier times, human shelters were mostly constructed from soil, and many containers were also fired from the soil. In modern life, soil is mainly used as an auxiliary material for reinforced concrete buildings. Even though humans built a large number of roads and hardened the roads in modern times [1], natural soil still covers a vast majority of the land's surface, such as suburbs, hills, forests, etc. In the design and application of protective structures, the focus is generally placed on the design and testing of new materials and structures, and soil is mainly used for protection against hazards such as large-caliber projectiles and explosive shock waves. There is little research on the damage of spherical, small-size penetrators similar to fragments. Fragments usually have a higher speed compared to large projectiles [2], so they pose a greater threat to organisms. In the wild or a relatively primitive natural environment, soil can usually be constructed by digging tunnels or stacking to construct simple protective structures. These facilities can be used for simple protection in experiments or to reduce the additional damage caused by high-speed fragments. In addition, due to the fact that soil is relatively easy to obtain and

can be quickly backfilled after damage, the overall damage to the structure is also relatively small, so it can be reused. Based on the above points, studying the protective ability of natural soil media against small-scale fragments is of practical value.

With the development and progress of science and the continuous development of metal materials, in order to deal with hidden fortifications and important facilities below the ground, countries have developed weapons and equipment represented by ground-penetrating bombs to penetrate deep soil and concrete protection. Therefore, studying the penetration of earth-penetrating projectiles into soil is quite important. Based on numerical simulation calculations, Zhou Yubing et al. [3] studied the effect of initial attack angle on the oblique penetration of earth-penetrating projectiles into soil, conducted numerical analysis of the penetration process at different angles, and studied the motion trajectory, penetration depth, and acceleration curve of the penetration process of the projectile. In response to the ballistic changes in projectile penetration into natural soil, Ren Baoxiang et al. [4] conducted ballistic tests and obtained trajectory images of projectile penetration into soil. It was found that penetration into soil would have a significant impact on the attitude of the projectile and further affect the depth of penetration. For different penetration depth calculation formulas, Yang Dongmei et al. [5] conducted numerical simulations on the penetration of a 57 mm-diameter kinetic energy projectile into a finite-thickness soil medium and concluded that the selected model is consistent with some soil types described by Sabsky and Berezin. Based on this type of research achievement, Jing Tong et al. [6] conducted experiments and numerical simulations on the penetration situation under low projectile velocity conditions and obtained a calculation method for the penetration depth and trajectory of ground-penetrating projectiles. From the above research results, it can be seen that the method of numerical simulation combined with experiments is more often applied to the process of projectiles penetrating the soil, with the diameter of the projectile studied by the above scholars being generally large and the projectile velocity being relatively low.

In order to gain a deeper understanding of the mechanism of projectile penetration into soil and select appropriate calculation models, Lin Jianxun [7] proposed the Spherical Cavity Expansion Theory, which considers the interface effects of different media for soil concrete composite media, and established a penetration model for projectile vertical penetration into soil concrete composite media. Xu Jianjun et al. [8] also analyzed the dynamic characteristics and motion laws of projectiles during penetration based on the Spherical Cavity Expansion Theory. On this basis, in order to study the effects of soil moisture content and dry density on projectile penetration resistance, Wu et al. [9] introduced these two conditions to modify the soil yield condition and completed the derivation of the Dynamic Spherical Cavity Expansion Theory. In order to further obtain the attenuation laws of different penetrators in soil, Gao Xingyong et al. [10] conducted impact compression performance experiments on soil materials with different parameters, obtained the stress and strain curves of the soil, analyzed the dynamic response characteristics of the materials based on their different characteristics, and obtained the influence of each parameter on the dynamic response.

In addition, many scholars have conducted extensive experiments and numerical simulations for different weapons and protective structures [11–17], proposed various numerical simulation calculation methods and parameter optimization methods, and analyzed various projectile penetration parameters. These scholars have proposed a variety of numerical simulation and parameter optimization methods and analyzed a variety of projectile penetration parameters. Based on these research results, there are many valuable references for the experimental design and application of numerical simulations in this field of study.

In order to obtain the characteristics and projectile parameters of prefabricated spherical fragments penetrating into soil, experiments with 6 mm- and 11 mm-diameter prefabricated fragments penetrating into soil were carried out in this paper. The purpose was to obtain a more accurate numerical calculation model of fragments penetrating into soil. Based on the relevant literature on the penetration of fast-moving fragments, the material



model suitable for this experiment was selected and compared with the experimental results [18–20]. In the second part, the target protection and parameters are analyzed and calculated based on these model parameters. In the third part, the fragment penetration effect under various initial conditions is studied. Combined with the actual protection structure, the protection parameters and characteristics of the soil medium are analyzed, which provides a reference for the design of relevant protection parameters.

## 2. Experiments on Fragment Penetration into Soil

### 2.1. Experimental Design Scheme

In the experiment, a 12.7 mm caliber ballistic gun was used as the launch carrier for the prefabricated spherical fragments, and the initial velocity of the prefabricated fragments was changed by changing the charge quality in the cartridge. After the prefabricated fragments are fired by the ballistic gun, they will pass through the velocity target in front of the target box (measuring the initial velocity before the fragments penetrate), the medium filling box filled with soil (the front and rear holes of the box do not block the fragments' penetration trajectory), the velocity target behind the target box (measuring the residual velocity after the fragments penetrate), and finally reach the collection box for the prefabricated fragments. The wires for the front and rear velocity targets are connected to the velocity measuring device. The schematic diagram of the experimental test set-up is shown in Figure 1.

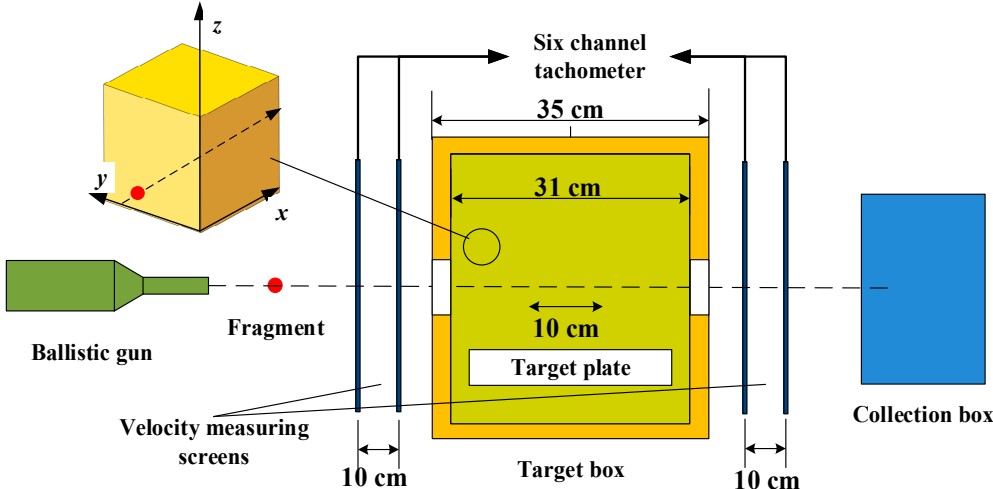

**Figure 1.** Schematic diagram of an experimental set-up for prefabricated fragment penetration into soil.

### 2.2. Experimental Operation Equipment

Ballistic gun tests were carried out on 11 mm and 6 mm tungsten alloy spherical fragments using the above device. In Figure 2a, ① is a fragment; ② is the sabot; and ③ is the cartridge. In Figure 2b, ① is the front velocity target; ② is the rear velocity target; ③ is a box for placing soil medium; and ④ is a six-channel speed measurement device (shown in Figure 3) that connects to the conductor of the tachometer. The test speed measurement adopts the on-off test method. The target paper is connected with the wire through the alligator clip, and the six-channel velocimeter is connected with the four target papers in front of and behind the target. When the fragments pass through the two target papers in front of the target box, the velocimeter can measure the time between the two target papers so as to obtain the initial velocity of the fragments. When fragments pass through two pieces of target paper behind the target box, the velocimeter can measure the time between the two pieces of target paper so as to obtain the residual velocity of the fragments.

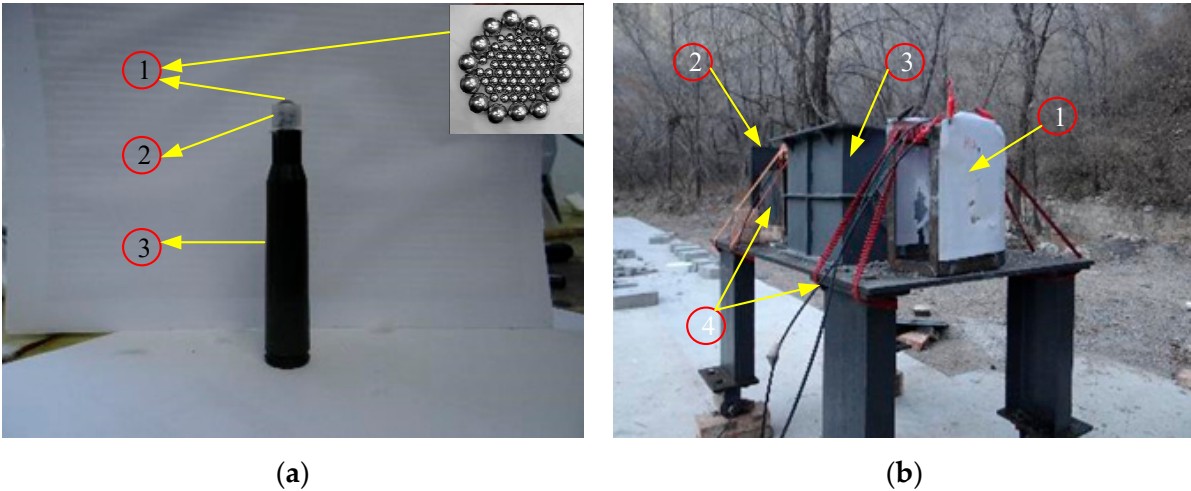

**Figure 2.** Physical diagram of the experimental set-up. (**a**) is the launched projectile structure; (**b**) is the experimental testing equipment.

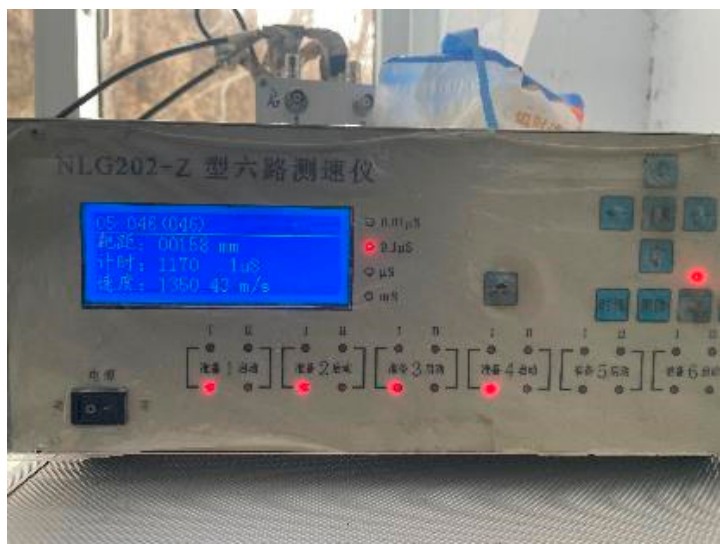

**Figure 3.** Six channel tachometer.

### 2.3. Experimental Results

From Table 1, it can be seen that 8 tests were conducted using 6 mm and 11 mm prefabricated tungsten alloy fragments in this experiment, with speeds ranging from 689 m/s to 1761 m/s (6 mm diameter) and 596 m/s to 1325 m/s (11 mm diameter). Fragments with a diameter of 6 mm cannot penetrate soil media with a thickness of 300 mm, while fragments with a diameter of 11 mm can penetrate that soil media. Figure 4 shows the front target and behind target results of fragments penetrating the soil medium in the experiment.

From Figure 4, it can be seen that on the front and rear target papers, in addition to large holes created when the fragments passed through, there is also a dense grouping of small holes left by sand splashing in the target box. Due to the looseness of soil as a medium, the trajectory left by fragments penetrating the soil medium under the influence of soil gravity is quickly buried. Therefore, the next main focus is on studying the velocity changes in prefabricated fragments.

**Table 1.** Experimental conditions and results of prefabricated fragments penetrating soil.

| No. | Fragment Diameter (mm) | Type of Target Plate | Speed Measurement in Front of Target (m/s) | Speed Measurement Behind Target (m/s) | Ability to Penetrate |
|---|---|---|---|---|---|
| 1 | 6 | Soil | 1269.05 | 0 | Unable |
| 2 | 6 | Soil | 1144 | 0 | Unable |
| 3 | 6 | Soil | 1761 | 15.67 | Able |
| 4 | 6 | Soil | 1533.58 | 0 | Unable |
| 5 | 6 | Soil | 985.03 | 0 | Unable |
| 6 | 6 | Soil | 885.44 | 0 | Unable |
| 7 | 6 | Soil | 763.28 | 0 | Unable |
| 8 | 6 | Soil | 689.55 | 0 | Unable |
| 9 | 11 | Soil | 1060.85 | 157.69 | Able |
| 10 | 11 | Soil | 721.46 | 87.35 | Able |
| 11 | 11 | Soil | 1154.10 | 121.31 | Able |
| 12 | 11 | Soil | 1178.23 | 173.18 | Able |
| 13 | 11 | Soil | 902.86 | 75.04 | Able |
| 14 | 11 | Soil | 755.62 | 83.96 | Able |
| 15 | 11 | Soil | 596.23 | 121.94 | Able |
| 16 | 11 | Soil | 1325.50 | 398.68 | Able |

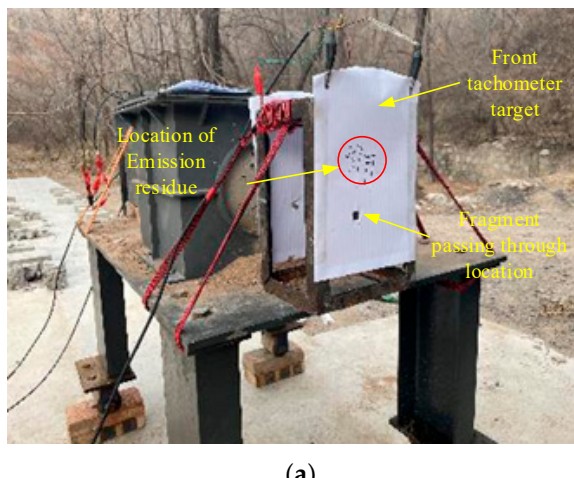

(**a**)

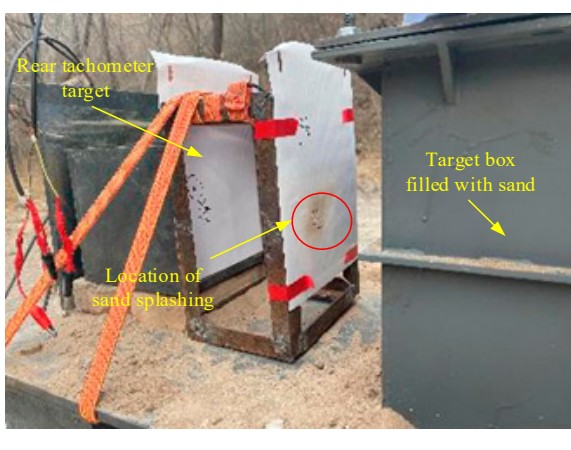

(**b**)

**Figure 4.** Results of fragment penetration into the soil medium in front of and behind the target.

## 3. Study on the Effect of Prefabricated Fragments Penetrating into the Soil Based on Numerical Simulation

In this part, in order to further study the mechanism of fragments penetrating into soil medium, firstly, the simulation model parameters of tungsten alloy and soil are obtained by using numerical modeling software based on the experimental results, so as to carry out the next part of the research and calculations. In practice, the motion parameters and initial conditions of fragments are relatively random, so it is necessary to simulate the penetration of fragments into soil at different speeds and angles. Combined with the common soil fortification forms in engineering, it is necessary to study under what conditions the protection conditions are effective, so a soil boundary condition analysis method is designed below, and the fragment motion parameters and protection parameters under this condition are obtained based on the simulation results.

In order to study the penetration effect of fragments penetrating soil under various conditions, the authors used LS-DYNA software to numerically simulate the penetration process of spherical tungsten alloy fragments. By comparing the residual velocity of fragments under the same initial conditions, the soil medium model parameters were modified. In the numerical simulation, the JC model [18–20] was used for tungsten alloy spherical

fragments, and the "Soil_and_Foam" (keyword name) model [4,6] was used for soil media. The detailed parameters of both models are shown in Tables 2 and 3, respectively.

**Table 2.** Keyword parameters of tungsten alloy materials.

| $R_0$ (g/cm$^3$) | $G$ (GPa) | $E$ (GPa) | $PR$ | $A$ | $B$ | $N$ | $C$ | $M$ | $TM$ (°C) | $TR$ (°C) | $EPSO$ |
|---|---|---|---|---|---|---|---|---|---|---|---|
| 17.5 | 137 | 350 | 0.22 | 1.51 | 0.177 | 0.12 | 0.016 | 1.0 | 1498 | 294 | 0.001 |

**Table 3.** Keyword parameters of soil materials.

| $R_0$ (g/cm$^3$) | $G$ (GPa) | $BULK$ (GPa) | $A_0$ | $VCR$ | $REF$ |
|---|---|---|---|---|---|
| 1.71 | 1.6 | 2.5 | $3.3 \cdot 10^{-4}$ | 0 | 1 |

In Table 2, $R_0$ is the material density; $G$ is the shear modulus; $E$ is the modulus of elasticity; $PR$ is Poisson's ratio; $A$, $B$, $N$, $C$, and $M$ are parameters of the relevant equation; $TM$ is the melting temperature; $TR$ is the room temperature; and $EPSO$ is the effective plastic strain rate. In Tables 3 and 4, $R_0$ represents the material density; $G$ is the shear modulus; $BULK$ is the bulk modulus; $A_0$ is the yield constant of the plastic yield function; $VCR$ is the volume crushing option; $REF$ is the initialization pressure using the reference geometry; $EPS_n$ is the volumetric strain value of the soil; and $P_n$ is the pressure value corresponding to the volumetric strain value.

**Table 4.** Volume strain values and pressure of soil (unit system: mm, ms, and kg).

| $EPS_1$ | $EPS_2$ | $EPS_3$ | $EPS_4$ | $EPS_5$ | $EPS_6$ | $EPS_7$ | $EPS_8$ | $EPS_9$ | $EPS_{10}$ |
|---|---|---|---|---|---|---|---|---|---|
| 0.00 | 0.05 | 0.09 | 0.11 | 0.15 | 0.19 | 0.21 | 0.22 | 0.25 | 0.30 |
| $P_1$ | $P_2$ | $P_3$ | $P_4$ | $P_5$ | $P_6$ | $P_7$ | $P_8$ | $P_9$ | $P_{10}$ |
| 0.00 | $4.5 \cdot 10^{-6}$ | $5.0 \cdot 10^{-6}$ | $6.7 \cdot 10^{-6}$ | $1.3 \cdot 10^{-5}$ | $2.1 \cdot 10^{-5}$ | $2.7 \cdot 10^{-5}$ | $3.9 \cdot 10^{-5}$ | 5.5 | 5.0 |

The "Soil_and_Foam" (keyword name) model was proposed by Krieg [21] in 1972, which can be used for soil and flattening foam, and has been well matched in the soil penetration test of high-speed projectiles by many scholars. The author used the same initial conditions as the experiment, with a fragment size of 11 mm and a soil medium size of 300 × 300 × 300 mm. All fragments were vertically shot into the soil medium. The comparison of the residual velocity results between the numerical simulation and the experiment is shown in Table 5.

**Table 5.** Comparison of residual velocities between the 11 mm fragment experiment and the numerical simulation results.

| No. | Initial Velocity (m/s) | Experimental Residual Velocity (m/s) | Simulation Residual Velocity (m/s) | Relative Error |
|---|---|---|---|---|
| 1 | 596.23 | 121.94 | 94.73 | 22.31% |
| 2 | 721.46 | 87.35 | 103.3 | 18.26% |
| 3 | 755.62 | 83.96 | 108.5 | 29.23% |
| 4 | 923.74 | 115.04 | 143.3 | 24.57% |
| 5 | 1060.85 | 157.69 | 174.2 | 10.47% |
| 6 | 1154.10 | 121.31 | 199.9 | 64.78% |
| 7 | 1178.23 | 173.18 | 200.8 | 15.95% |
| 8 | 1307.60 | 298.68 | 233 | 21.99% |

From Table 5, it can be seen that the numerical simulation results are more stable than the experimental results. This is mainly because the experimental conditions have a certain

degree of uncertainty, but the average error of the remaining velocity of fragments is 25.94%, and the numerical simulation results of fragments penetrating the soil have credibility.

Based on the above experimental and numerical simulation data, combined with the Poncelet formula, the residual velocity of fragments is determined. Equation (1) is a general expression of the Poncelet formula, which is characterized by a proportional relationship between the square of the initial velocity and the square of the residual velocity. Due to the limited available experimental data for fragments with a diameter of 6 mm, the author performed a formula fitting on fragments with a diameter of 11 mm, and the resulting fitting formula is shown in Equation (3).

$$v^2 = v_0{}^2 \exp(-\frac{2c_3 Ah}{m}) + \frac{c_1}{c_3} \exp(-\frac{2c_3 Ah}{m}) + 1 \tag{1}$$

$$v^2 = v_0{}^2 \exp(-\frac{c_3 h}{r\rho}) + \frac{c_1}{c_3} \exp(-\frac{c_3 h}{r\rho}) + 1 \tag{2}$$

$$v^2 = v_0{}^2 \exp(-\frac{3051.1h}{r\rho}) + \frac{1165239166.7}{3051.1} \exp(-\frac{3051.1h}{r\rho}) + 1 \tag{3}$$

$$v^2 = v_0{}^2 \exp(-\frac{3240.5h}{r\rho}) + \frac{636696645.8}{3240.5} \exp(-\frac{3240.5h}{r\rho}) + 1 \tag{4}$$

Based on the numerical simulation calculation model, the following research mainly focuses on two aspects: the angle of attack of fragment penetration and the boundary effect of finite-thickness soil.

### 3.1. Impact of the Initial Attack Angle of Fragments on Soil Penetration Depth

Based on the numerical model of fragment penetration into soil mentioned above, the authors observed the vertical penetration depth and trajectory penetration depth of the fragment from the surface by changing its initial incidence angle. The initial velocities of the fragments were selected as 500 m/s, 1000 m/s, and 1500 m/s. At each fragment velocity, a total of nine attack angles were conducted, including 0°, 10°, 20°, 30°, 40°, 50°, 60°, 70°, and 80°. The final penetration position of fragments in numerical simulation is shown in Figure 5.

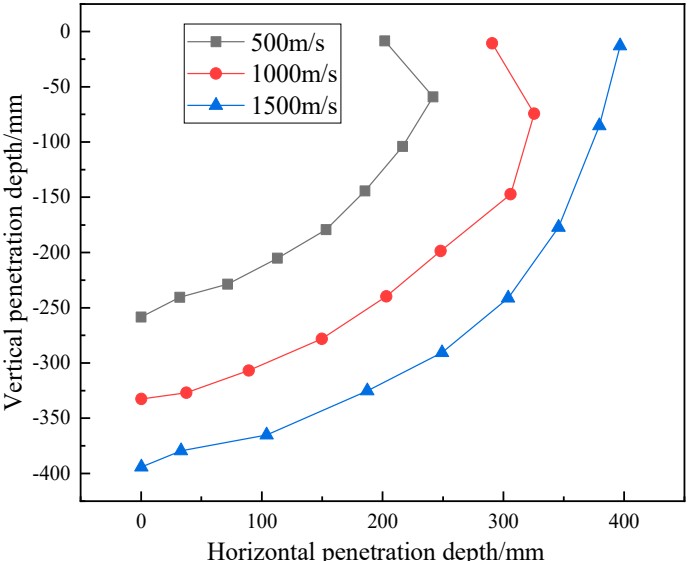

**Figure 5.** Different directional depths of spherical fragments penetrating soil at different angles of attack and velocities.

It can be seen from Figure 5 that with the increase in the initial incident angle of fragments, the absolute value of the penetration depth perpendicular to the soil surface gradually and linearly decreases from 0° to 30°. However, the depth value gradually increases after 30°. With the increase in the initial velocity of fragments, the maximum depth difference in the vertical direction is 91.94 mm (60°~70°) at 1500 m/s, and the depth in the horizontal direction is 83.4 mm (20°~30°). However, the length of the penetration trajectory at the end point first decreases and then increases under various velocity gradients, as shown in Figure 6.

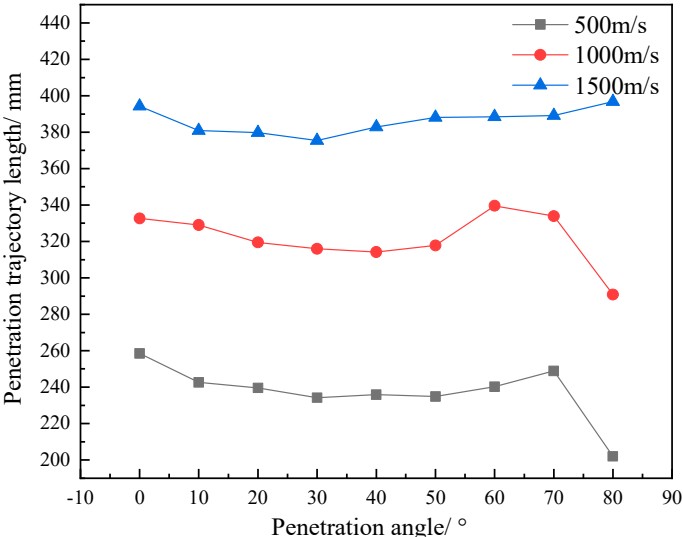

**Figure 6.** Penetration depth of fragments at different speeds based on variations in penetration trajectory length with initial angle of attack.

It can be seen from Figure 6 that the fragment trajectory length of different initial velocities reaches the minimum value between 30°~40° in the range of 0°~70°, indicating that the soil has a greater impact on fragment penetration at this time. The main reason for the reduction in the vertical depth and penetration trajectory depth of fragments penetrating into the soil is that the angle of attack reduces the vertical partial velocity and causes the fragments to approach the surface area of the soil. From Figure 7, the stress state of the soil on fragments under the penetration trajectory at different angles can be seen. With the increase in the angle of attack, the angle between the force direction of the fragment at the bottom and the velocity direction of the fragment increases, resulting in a gradual increase in the trajectory offset of the fragment, which affects the penetration ability of the fragment into the soil.

The deviation angle is mainly related to the initial attack angle of the fragments. Figure 8 shows the variation in fragment deviation angle with attack angle under three velocity gradients. Regardless of the initial velocity of fragments, the variation trend of the deviation angle of fragments has similar regularity. As the initial angle of attack increases, the vertical velocity of fragments decreases rapidly while the horizontal velocity increases. With the increase in the angle of attack, the thickness of the soil above the fragment in the process of penetration gradually decreases. This is affected by the free interface effect on the upper surface of the soil medium, causing the upper and lower parts of the fragment to be subjected to different sizes of stress. Consequently, there is a gradual increase in the deflection angle in the process of fragment movement. The speed reduction in the horizontal direction is relatively low. It can be seen from Figure 8 that the deviation angle of fragments is negative in the range of 0°~40° and reaches its maximum value at 20°. The deviation angle of fragments is positive in the range of 40°~80°, and the maximum value is obtained near 80°.

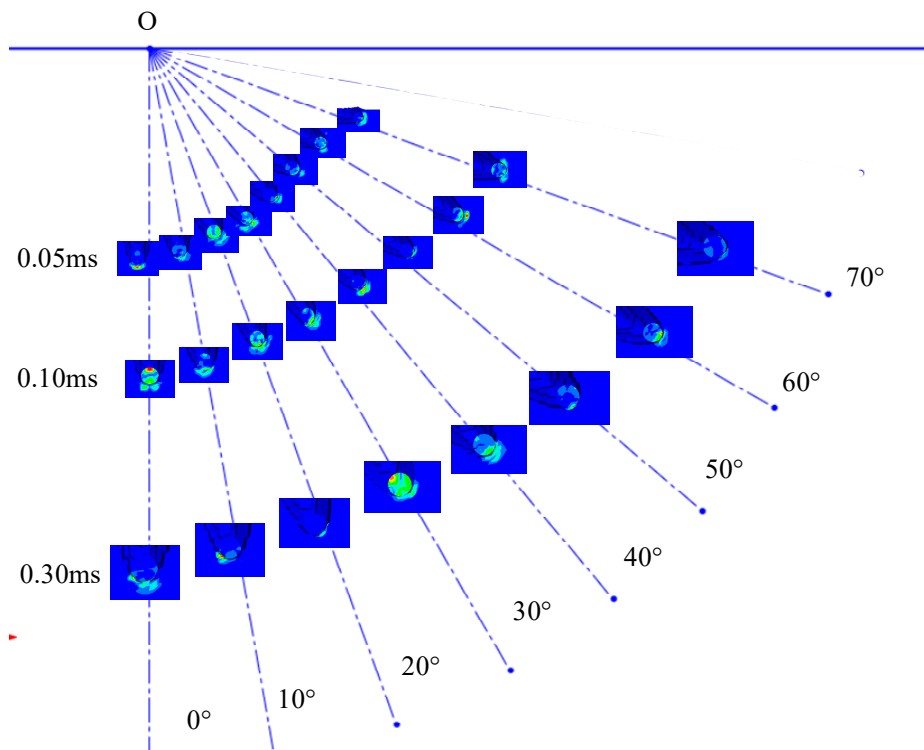

**Figure 7.** Stress cloud map of a fragment moving at 1000 m/s at different angles of attack.

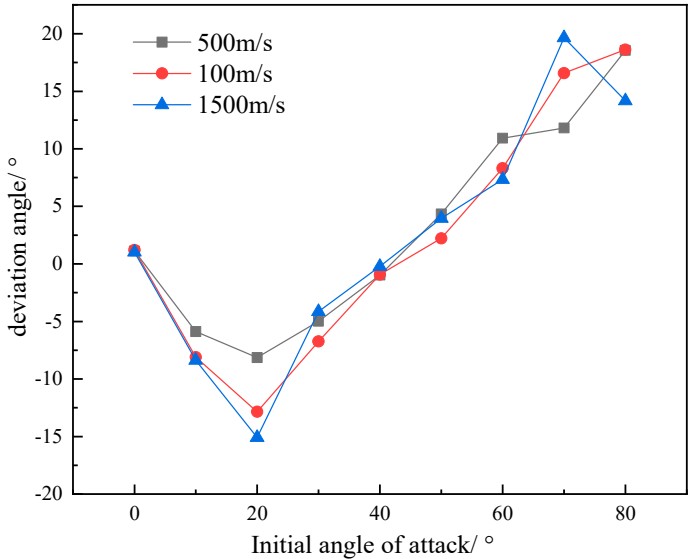

**Figure 8.** Deviation angle of fragments with different velocities as a function of attack angle.

### 3.2. The Influence of Boundary Effects on Fragment Penetration on Finite Thick Soil of Different Trajectories

This part of the content is based on practical considerations. As shown in Figure 9, there are two hypothetical scenarios: (a) is the simulation of the protective effect of sandbags on personnel, and (b) is the simulation of the protective effect of bunkers on personnel. Based on the similarity between the two scenarios, the sandbags and bunkers are equivalent to a rectangular boundary block of soil. The equivalent diagram of sandbags and bunkers is shown in Figure 9.

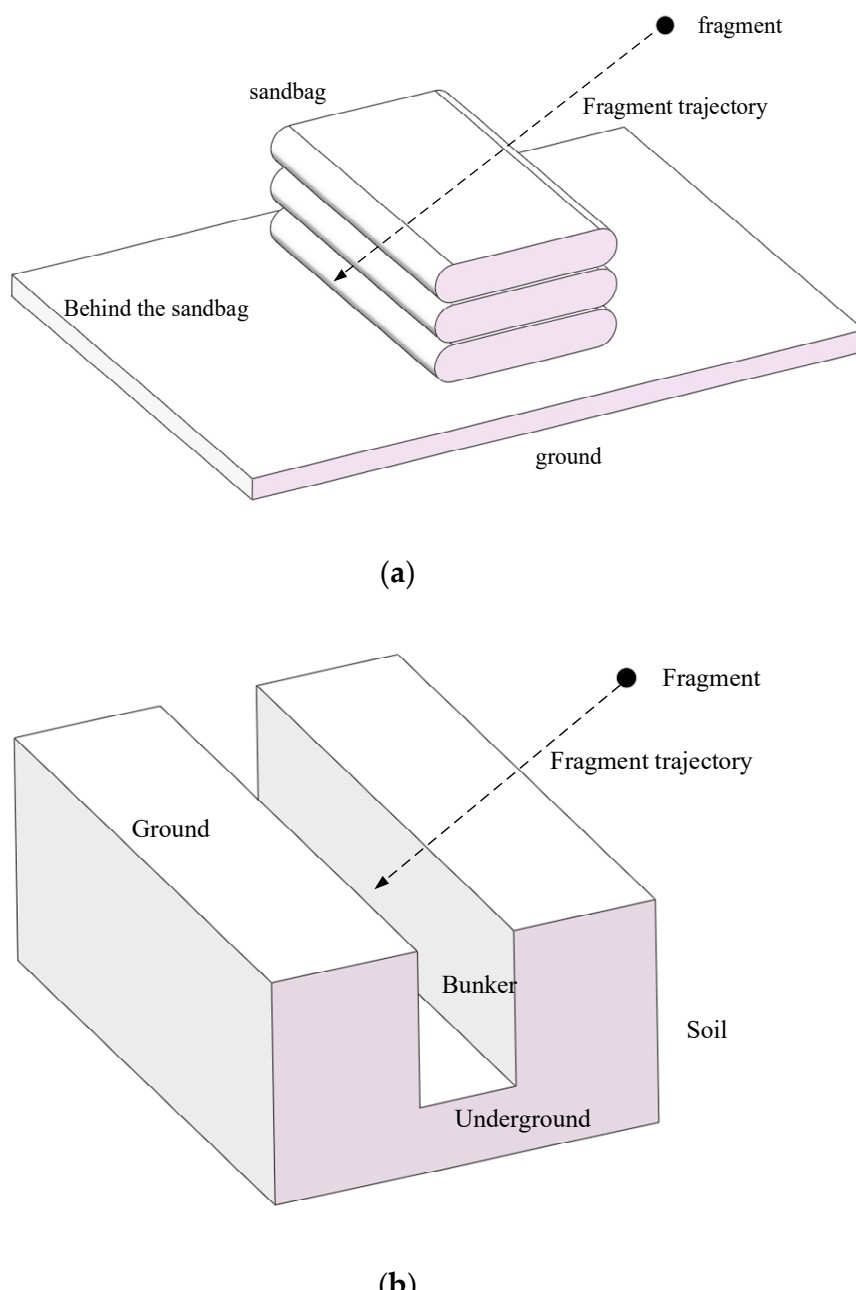

**Figure 9.** Illustrated schematic diagram of two scenarios. (**a**) is a schematic diagram of sandbag protection; (**b**) is a schematic diagram of bunker protection.

From Figure 9, it can be seen that when determining the source of fragments in a certain direction, people located behind sandbags and in bunkers can only consider the protective effect on one side (the other side is the same). The soil parts located above and to the left are exposed areas that are easily affected by soil-free boundary effects, while the lower and right sides of the soil are non-exposed infinite areas that are less affected by the effects of free interfaces. Based on this assumption, the authors conducted numerical simulation calculations on the penetration of prefabricated fragments with different initial conditions into the soil. The intersection point of two free interfaces was selected as the origin, and the distances $X_1$ and $Y_1$ from the origin O were selected as the entry and exit points of the fragments. The initial velocity of the fragments was 1500 m/s. The working conditions of the numerical simulation are shown in Figure 10.

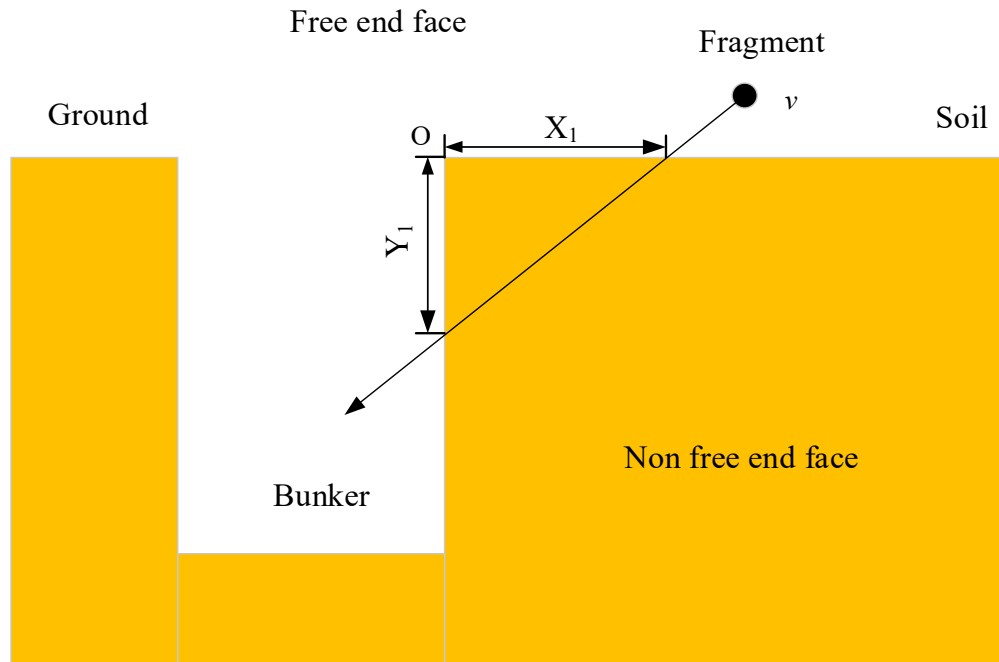

**Figure 10.** Diagram of boundary effect conditions for fragments penetrating soil media.

In Table 6, the length of the penetration trajectory is used as the standard for measuring the remaining velocity of prefabricated fragments. If there is a clear functional relationship between the residual velocity and the length of the fragment penetration trajectory, but there is no significant relationship between it and the distances $X_1$ and $Y_1$ from the origin, it can be concluded that the soil does not have a significant boundary effect. Figure 11 shows the three-dimensional relationship between the absolute difference in residual velocity of fragments and the boundary distances $X_1$ and $Y_1$.

**Table 6.** The residual velocities of boundary effects and a comparison of fragments penetrating soil media.

| No. | $X_1$ (mm) | $Y_1$ (mm) | Length of the Penetration Trajectory (mm) | Residual Velocities of Fragments (m/s) | Comparison of Residual Velocities (m/s) | Proportion of Differences (%) | Speed Difference (m/s) |
|---|---|---|---|---|---|---|---|
| 11 | 25 | 25 | 35.4 | 1294 | 1238 | 4.52 | 56 |
| 12 | 25 | 50 | 55.9 | 1145 | 1111 | 3.06 | 34 |
| 13 | 25 | 75 | 79.1 | 997.4 | 983.2 | 1.44 | 14.2 |
| 14 | 25 | 100 | 103.1 | 862.4 | 867.1 | 0.54 | 4.7 |
| 21 | 50 | 25 | 55.9 | 1189 | 1111 | 7.02 | 78 |
| 22 | 50 | 50 | 70.7 | 1076 | 1028 | 4.67 | 48 |
| 23 | 50 | 75 | 90.1 | 985.5 | 928.7 | 6.12 | 56.8 |
| 24 | 50 | 100 | 111.8 | 843.7 | 825 | 2.27 | 18.7 |
| 31 | 75 | 25 | 79.1 | 1075 | 983.2 | 9.34 | 91.8 |
| 32 | 75 | 50 | 90.1 | 982.2 | 928.7 | 5.76 | 53.5 |
| 33 | 75 | 75 | 106.1 | 886.9 | 852.7 | 4.01 | 34.2 |
| 34 | 75 | 100 | 125 | 797.8 | 770.7 | 3.52 | 27.1 |
| 41 | 100 | 25 | 103.1 | 971.1 | 867.1 | 12.06 | 104 |
| 42 | 100 | 50 | 111.8 | 875.5 | 825 | 6.12 | 50.5 |
| 43 | 100 | 75 | 125 | 810.9 | 770.7 | 5.22 | 40.2 |
| 44 | 100 | 100 | 141.4 | 726.5 | 702.6 | 3.40 | 23.9 |

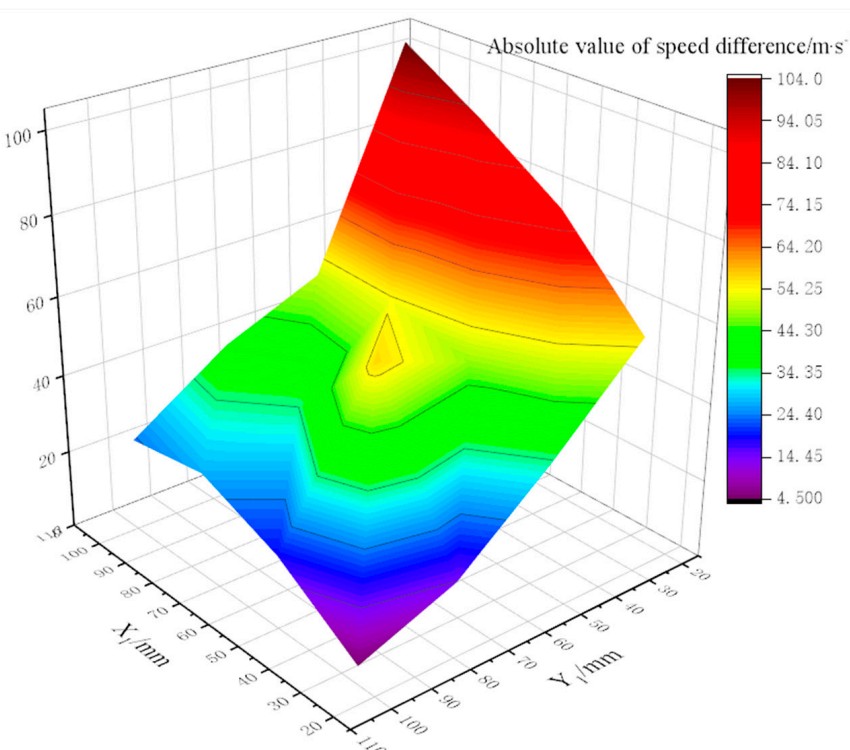

**Figure 11.** Relationship between the difference in residual velocity of fragments and boundary distances $X_1$ and $Y_1$.

It can be seen from Figure 10 that although the residual velocity of fragments has a relatively obvious inverse proportional relationship with the length of the penetration trajectory as a whole, we have noticed that within the small value range of $X_1$ and $Y_1$, the reduction in fragment velocity is relatively small. It is obvious that this is because it is close to the free boundary of the soil, which affects the structural strength of this part of the soil and reduces the resistance of fragments penetrating the soil, resulting in this situation. Therefore, it can be seen from Figure 10 that under the initial condition of 1000 m/s, with the increase in $X_1$, the influence of boundary action gradually increases, and the growth rate slows down after 70 mm. The smaller the $Y_1$, the more obvious the effect. With the increase in $Y_1$, the influence degree of the boundary effect gradually decreases, and the influence difference between the two kinds of boundaries after 80 mm is reduced to less than 20 mm, indicating that the soil boundary effect will not be obvious after that.

In addition to the areas studied above that are more prone to soil boundary effects, it is also necessary to determine which areas are relatively safe from the perspective of protective engineering. The residual velocity of fragment damage elements is used as the basis for measuring their killing power. Numerical simulations of fragment incidence are carried out at four distances of 50 mm, 150 mm, 250 mm, and 350 mm, and the calculated results of fragment residual velocities are shown in Table 7.

From Table 7, it can be seen that as the incident and exit positions of fragments continue to move away from the original point O, the residual velocity of fragments decreases significantly, and the effective penetration depth of 1500 m/s fragments into soil media is approximately 400 mm. It can be seen from Figure 12 that with the increase in horizontal distance, the residual velocity of fragments shows a decreasing trend as a whole. However, when the vertical distance is too small, the residual velocity of fragments changes due to the influence of the boundary. When the vertical distance is within 100 mm, the residual velocity of fragments first decreases and then increases. In contrast, with the increase in vertical distance, the residual velocity of fragments shows an obvious decreasing trend as a whole.

**Table 7.** Results of fragment penetration into soil under different flying conditions.

| No. | Length from Point O in a Horizontal State (mm) | Length from Point O in a Vertical State (mm) | Length of the Penetration Path (mm) | Angle of Incidence (°) | Residual Velocity (m/s) |
| --- | --- | --- | --- | --- | --- |
| 11 | 50 | 50 | None | 45 | 1015 |
| 12 | 50 | 150 | None | 18.43 | 500.7 |
| 13 | 50 | 250 | None | 11.31 | 104.9 |
| 14 | 50 | 350 | 376.5 | 8.13 | 0 |
| 21 | 150 | 50 | None | 71.56 | 585.3 |
| 22 | 150 | 150 | None | 45 | 373.5 |
| 23 | 150 | 250 | None | 30.96 | 184.4 |
| 24 | 150 | 350 | 367.9 | 23.20 | 0 |
| 31 | 250 | 50 | None | 78.69 | 345.7 |
| 32 | 250 | 150 | None | 59.04 | 194.9 |
| 33 | 250 | 250 | None | 45 | 94.7 |
| 34 | 250 | 350 | 376.5 | 35.53 | 0 |
| 41 | 350 | 50 | Ricochet | 81.87 | 699.2 |
| 42 | 350 | 150 | None | 66.80 | 76.65 |
| 43 | 350 | 250 | 371.4 | 54.46 | 0 |
| 44 | 350 | 350 | 378.9 | 45 | 0 |

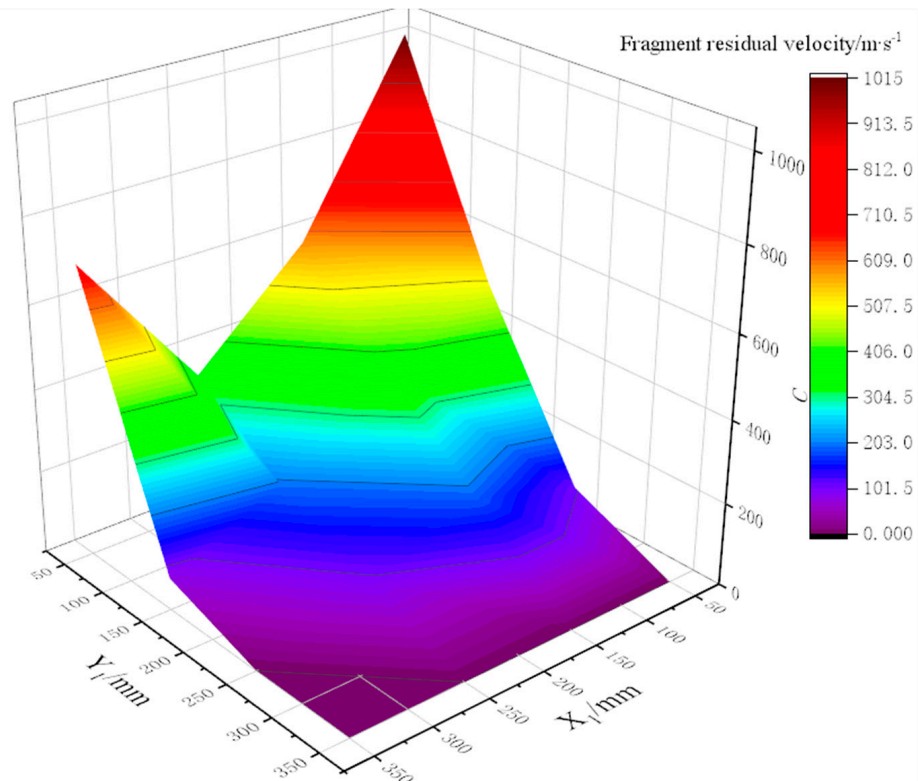

**Figure 12.** Changes in residual velocity of fragments with horizontal and vertical distances.

It can be seen from Figure 13 that with the increase in the initial velocity of fragments, the penetration depth of fragments increases and gradually slows down, and it is difficult for fragments to continue to penetrate after 500 mm. Taking the maximum speed of 2000 m/s as the standard, it can be concluded that a soil medium more than 500 mm away from the upper surface of the trench can achieve a useful protective effect.

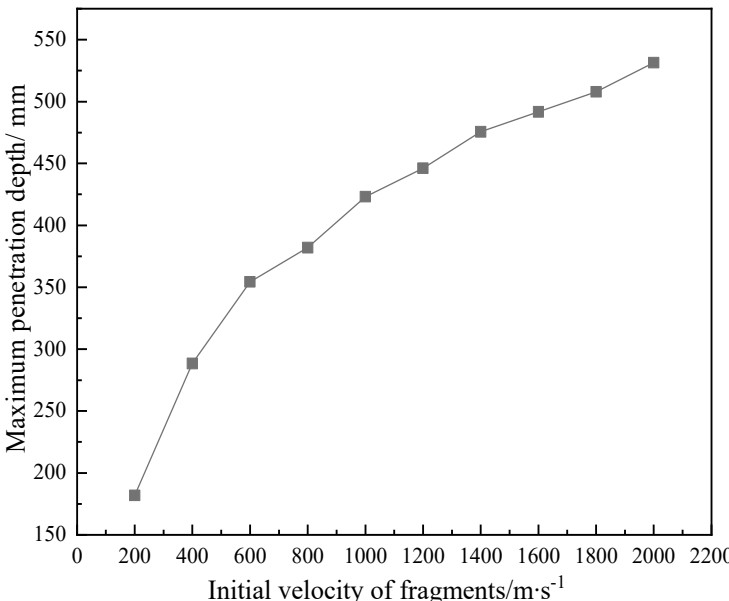

**Figure 13.** Maximum penetration depth of fragments at different velocities.

## 4. Conclusions

Based on the penetration test of prefabricated tungsten alloy spherical fragments into soil media, the authors analyzed the experimental phenomena and results of fragments with diameters of 6 mm and 11 mm penetrating soil and obtained the residual velocities of fragments at different initial velocities. Based on the parameters of the soil numerical calculation model, the authors also calculated the penetration force of spherical fragments with different attack angles, velocities, and target thicknesses on the soil and analyzed the calculation method for the residual velocity of fragments entering the soil. In addition, the authors analyzed the motion parameters and stress states of fragments penetrating soil media from two aspects: the initial velocity and angle of attack, and obtained the general laws of fragment motion at different angles of attack ($0°\sim80°$). On this basis, combined with the relevant initial conditions that are mathematically equivalent to the actual situation of the two free surfaces of the tunnel, the authors analyzed the influence of the boundary effect of small (25 mm~100 mm) soil medium on the penetration process of fragments and compared the penetration results of large (50 mm~350 mm) soil medium. Finally, the authors analyzed the maximum penetration depth under different velocity gradients.

(1) Based on the analysis method of the Poncelet formula and the experimental data, the formula for calculating the residual velocity of prefabricated tungsten alloy spherical fragments penetrating into soil medium under a finite thickness is obtained, and the effective protective thickness of soil can be roughly calculated through the formula.

(2) The initial attack angle of fragments has a great influence on the length and deviation angle of the penetration trajectory. The essence of this experiment is to change the distance between fragments and the free surface during the penetration process, which affects the stress state of fragments and results in a change in the penetration motion parameters of fragments and changes in the deviation direction of fragments with a boundery of $40°$.

(3) Through the established mathematical model analysis method, the authors determined that the main influence range of boundary effects under small-scale conditions is 0 mm~80 mm, which shows different changing trends in two directions. Under large-scale conditions, the residual velocity of fragments decreases with an increase in penetration length, but when the vertical distance is too small, the residual velocity of fragments will increase. When the soil thickness is greater than 500 mm, it can be considered that a safe protective effect has been achieved.

By studying the effectiveness of soil against fragment penetration, this article can provide a reference for the design of soil protection parameters in natural environments. Through experiments, it can be seen that in-depth research on the protective effect of the initial angle of attack on finite-thickness targets is of great significance. The research method for boundary effects of soil blocks in the study can be extended to other types of protective structure design, and the influence of penetrating body parameters can be considered, thus establishing the connection between free surface effects and final motion parameters.

**Author Contributions:** Conceptualization, Z.W. (Zhenning Wang) and J.Y. (Jianping Yin); Methodology, Z.W. (Zhenning Wang); Software, Z.W. (Zhenning Wang); Validation, Z.W. (Zhenning Wang); Investigation, Z.W. (Zhenning Wang), J.Y. (Jianya Yi), X.L. and J.Y. (Jianping Yin); Experiment, Z.W. (Zhijun Wang) and X.L.; Data curation, Z.W. (Zhenning Wang) and J.Y. (Jianya Yi); Writing—original draft preparation, Z.W. (Zhenning Wang); Writing—review and editing, J.Y. (Jianping Yin); Supervision, X.L. and J.Y. (Jianping Yin); Project administration, Z.W. (Zhijun Wang). All authors have read and agreed to the published version of the manuscript.

**Funding:** The authors would like to acknowledge the financial support from the 2022 Basic Research Program of Shanxi Province (Free Exploration), grant number 202203021212136.

**Institutional Review Board Statement:** Not applicable.

**Informed Consent Statement:** Not applicable.

**Data Availability Statement:** The data presented in this study are available on request from the corresponding author. The data are not publicly available due to programming privacy in structural design.

**Conflicts of Interest:** The authors declare no conflict of interest.

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
