# Peer review of "Impact of Soil on the Penetration of Prefabricated Spherical Fragments and Its Protective Effect"

_applsci, doi:10.3390/app132011567_

Round 1

Reviewer 1 Report

The paper is well-written and the study is well-presented. The quality of figures is high. the results and the conclusions are appropriate. Only a light editing is requested. In detail, please revise the first sentence of the abstract where a repetition is present. Also, revise the position of table 6 which is at the moment on two pages and finally check the format of the references. 

Author Response

Please refer the attachment.

Reviewer 2 Report

1.   The abstract section needs more clarification regarding the soil condition and its results to be quantified.

 2.      The originality of the paper needs to be stated clearly. It is important to have sufficient results to justify the novelty of a high-quality journal paper. The Introduction should make a compelling case for why the study is useful along with a clear statement of its novelty or originality by providing relevant information and providing some basic relevant information.

3. The discussion is not up to the mark in the angle of attack of fragment penetration and the boundary effect of finite thickness soil

4. Numerical simulations of fragment incidence are 251 carried out at four distances of 50mm, 150mm, 250mm, and 350mm, on what basis you selected these four distances?
5.      English grammar must be checked by a native English speaker.

6.      The conclusion should not just be a summary of the work, but more about limitations and future works should be added. And also quantify the soil thickness results.

7.  Updated the references 

1.   The abstract section needs more clarification regarding the soil condition and its results to be quantified.

 2.      The originality of the paper needs to be stated clearly. It is important to have sufficient results to justify the novelty of a high-quality journal paper. The Introduction should make a compelling case for why the study is useful along with a clear statement of its novelty or originality by providing relevant information and providing some basic relevant information.

3. The discussion is not up to the mark in the angle of attack of fragment penetration and the boundary effect of finite thickness soil

4. Numerical simulations of fragment incidence are 251 carried out at four distances of 50mm, 150mm, 250mm, and 350mm, on what basis you selected these four distances?
5.      English grammar must be checked by a native English speaker.

6.      The conclusion should not just be a summary of the work, but more about limitations and future works should be added. And also quantify the soil thickness results.

7.  Updated the references 

Author Response

Please refer the attachment.

Author Response

Please refer the attachment.

Round 2

Reviewer 3 Report

The draft has been significantly implemented compared to the initial version. However, when reading the text, there are still inaccuracies and errors (oversights), especially in punctuation, for example after a comma there must not be a capital letter, it only goes after the period. In tables 2,3 and 4 I suggested putting the units for easier reading of the parameters, adding "Table 3. Keyword Parameters of soil materials(Unit system: mm; ms; kg)" in itself does not help the reader to understand immediately what he is reading, but will necessarily have to continually read what is reported in the caption of the figure. My suggestions (these as well as others) were only intended to improve the reading and give greater readability to the great work done by the authors.
